# A Review of the Financial Value of Faecal Sludge Reuse in Low-Income Countries

**Adrian Mallory** [1], **Rochelle Holm** [2] **and Alison Parker** [1,*]

1   Cranfield Water Science Institute, Cranfield University, Cranfield MK43 0AL, UK; a.mallory@cranfield.ac.uk
2   Centre of Excellence in Water and Sanitation, Mzuzu University, P/Bag 201, Mzuzu 2, Malawi;
    rochelle@rochelleholm.com
*   Correspondence: a.parker@cranfield.ac.uk

**Abstract:** Faecal sludge reuse could promote responsible waste management and alleviate resource shortages. However, for this reuse to be carried out at scale, it needs to be financially viable. This paper reviews the financial values of resource recovery from 112 data points from 43 publications from academic and grey literature. The results found 65% of the existing literature is projected rather than being based on observed data from products in practice, with limited studies providing actual experiences of revenue in practice. Some of the estimates of the potential value were ten times those observed in data from operating businesses. Reasons for this include pricing of products against unrealistic competitors, for example, pricing briquettes against diesel fuel, or difficulties in marketing or regulation of products in practice. The most common form of reuse in practice is agricultural composting, which is also the lowest value product. Few cases were able to achieve more than $5/person/year from sludge reuse, therefore other drivers are needed to promote proper human waste disposal, including the health and dignity of citizens, but which are not easily monetised. Certification and recognition of product safety can improve the perception of value and products. Resource recovery has a limited role in the financial viability of providing Circular Economy sanitation in low-income countries. Instead, there is a need to focus on supportive policies and subsidies enabling the transition towards a Circular Economy supporting environmental quality, ecological health and human health.

**Keywords:** agriculture; economics; faecal sludge management; reuse; sanitation; waste management

## 1. Introduction

Providing safe sanitation in low-income countries and achieving Sustainable Development Goal (SDG) number six to "ensure availability and sustainable management of water and sanitation for all" [1] remains a major hurdle to 61% of the global population. Low-income countries are often served by onsite technologies, such as pit latrines, with 5 billion people expected to be served by onsite sanitation by 2030. In addition, faecal sludge (FS) treatment plants often fail after construction due to lack of ongoing finances for operations [2]. Failures in waste management are contaminating natural resources, which are also being stressed by the extraction of raw materials. The lack of financing for sanitation and increased contamination of natural resources from inappropriate disposal of waste has led to an increased advocacy for a shift to Circular Economy (CE) approaches to sanitation which can yield nutrients for agriculture, protein for animal feed or clean energy [3,4]. Revenues from these products could fund the maintenance of sanitation systems whilst providing an incentive for proper waste management and preventing pollution [4].

CE sanitation has many institutional arrangements: operated by social enterprises [5–7], led by local or national governments [8–10] or informally practiced by community members [11–13]. Despite the

advocacy for the potential of CE sanitation, revenues from the sale of products are often theoretical projections, meaning the actual potential to fund sanitation is not yet known. The current evidence of the potential to scale and achieve such revenues is fragmented and context specific [14]. Individual findings or cases of FS reuse could succeed or fail for many contextual reasons, so there is a need to understand the potential across many cases. There is limited data about the costs of sanitation in general [15], meaning the potential of different revenue streams to fund sanitation provision is not understood. As business models in the arena of FS reuse are often young and nascent, determinants of value such as scale, input waste, marketing, political recognition and certification are not well understood. There are three major gaps in knowledge about CE sanitation: the value of FS reuse in theory and practice, the determinants that affect these values and the amount that FS reuse can contribute to funding sanitation. As more businesses, NGOs and governments have started practicing CE [8], there is more data available about the value of FS re-use in practice. This paper aims to address the following questions:

1.　What is the financial value of FS re-use?
2.　Is there a gap between theoretical value projections and the actual value that can be recovered?
3.　What determines the value of FS reuse?
4.　Can FS re-use act as a driver for improved sanitation?

This paper reviews existing case studies of FS reuse in low-income countries and looks at the revenue being generated from reuse of FS to answer these four questions. Papers predicting or projecting potential values of FS reuse are also reviewed to compare the difference between theoretical propositions and actual value.

## 2. Materials and Methods

To investigate the financial value of FS recovery, a literature review was chosen as the most suitable method. This allows the research to capture and summarise values from small studies in different contexts and with different models of resource recovery. This includes historical projects practicing FS reuse that may no longer be operational. The final dataset and analysis makes a larger contribution to knowledge than the current state of literature of multiple single-case studies that are not compared or contrasted with each other, forming a significant contribution to knowledge to inform future studies of FS reuse.

### 2.1. Data Collection

A dataset of values of FS was collected from any papers found that stated a financial value of re-using FS. An initial focused group of key documents was identified using the knowledge hub of the sustainable sanitation alliance (SuSanA). SuSanA is a network for individuals and organisations working in sanitation; a combination of publication hub, grey literature and discussion forum providing the latest state of knowledge of the sector [16]. Searches were additionally performed within Scopus, Google Scholar and the Water Engineering and Development Centre at Loughborough University knowledge—a forum holding conference papers, publications and reports for policymakers and practitioners in the water and sanitation sector [17]. Terms used included 'sanitation as a business', 'financial value excreta', 'circular economy sanitation', 'faecal sludge reuse', 'resource recovery sanitation' and 'excreta resource recovery'. The review process only considered examples from countries defined as low or middle income by the World Bank [18]. Papers were filtered by whether or not they presented any financial data about the FS reuse potential in low-income countries. Papers presenting financial data were reviewed and are summarised in Table 1. The upstream technology used (sewers, Container-Based Sanitation [CBS], septic tanks or pit latrines) was also recorded to see if it influenced the value of the product recovered. The majority of the cases looked at FS either from on-site sanitation, defined as systems where the excreta is stored on the plot where generated, such as pit latrines, or septic tanks [19]. Of these, 45% of the papers surveyed were published in 2015 or later

and 83% had been published since 2010. Where businesses practising resource recovery were known but financial data about reuse could not be found, or needed clarification, email communication was used to seek further information. This was done for two organisations.

**Table 1.** Types of Faecal Sludge Resource Recovery Studied.

| Faecal Sludge Resource Type | Number of Values Considered | References |
| --- | --- | --- |
| Fuel (Briquettes) | 14 * | [9,10,14,20–25] |
| Biogas | 18 | [8,14,20,25–33] |
| Compost | 53 | [6,8,11,13,14,20,21,23,25,32–54] |
| Protein feed (Black Soldier Fly) | 8 | [14,21,22,25] |
| Urine reuse in agriculture | 8 | [8,20,32,36,47,50,51,55] |
| Aquaculture (Plants or fish from treatment ponds) | 5 | [8,26,56] |
| Recovered water | 4 | [8,20,57] |
| Building material | 1 | [8] |

* In some cases, multiple values are in a single piece of literature.

## 2.2. Data Analysis

As the aim was to understand the financial potential of FS reuse, and data on both capital and operational costs were limited; the main data point selected was the sale price of excreta-derived products to an end user. Any study detailing a value or revenue derived from reuse was analysed in the study. Only one paper had detailed analysis of both the costs of providing sanitation services and revenue from resource sales [34]. Whilst this fails to account for the whole business feasibility, it offers a view into the scale of revenue collection potential and can be compared to the actual potential of CE sanitation.

Data were categorised according to: input (sewers/FS/container-based sanitation), location and resource produced. Data about revenues from FS reuse were in three categories: total revenue per year with no reference point for volume collected or people served, revenue per year per person served and revenue per year per volume collected/treated. The most common category of revenue data available was in revenue per person per year. Diener presented values from three different cities in tonnes and volume of FS, giving a density value to convert values between volumes and mass [14]. Dodane [28] gave an estimated FS production of 2.7L/p/d, giving an estimated volume per population. These conversion factors were used to convert literature values to $/p/y, where the original values were in $/m$^3$/y or $/t/y. This meant all data could be converted into comparable values. FS is a variable material [2,19]; thus, relying on such assumptions of uniform density or FS production per person is a methodological limitation. Values were also converted from local currency to dollars using the World Bank exchange rate [58] at year of publication. The dollar value was then adjusted to 2019 for inflation. The process of data collection and conversion resulted in 109 data points from 43 publications.

To analyse the difference between the theoretical potential for FS recovery in the literature and actual prices, data were assigned a 'score': the higher the score, the more theoretical the revenue was. This was done by breaking down the method into three sections: volume of waste collected, volume/quality of product and sales price. Four categories were given a binary score of 0 to 3 (Table 2). For example, for businesses actively selling products, the financial data was based on observed collection, production and sales, resulting in a 'theory score' of 0. The results are available as Supplementary Materials as detailed in the appendix. Value projections based on collecting all sludge from the population of a large city were given a theory score of 3. This holistic approach provided a metric for viewing the difference between theory and practice.

**Table 2.** Scores for Available Literature.

| Score | Criteria | Literature |
|:---:|:---:|:---:|
| 0 | Values from active businesses: observed collection, production and prices | [6,8,11,13,23,28,31–35,37,38,43,51,53,54,56,57] |
| 1 | Literature or assumption used to estimate potential for one of: collection, production or price of product | [9,10,23,24,32,35,36,41,42,45,47,49,52,59] |
| 2 | Literature or assumption used to estimate potential for two of: collection, production or price of product | [14,25–27,29,32,46,48,50–52,55] |
| 3 | Potential value projection: literature and assumptions used to estimate collection, production and price of product | [20–22,25,40,60] |

*2.3. Thematic Analysis*

The drivers of reuse revenues were also explored. This was done by reviewing the literature behind the financial data, informal interviews and discussions with businesses in resource recovery associated with the literature reviewed and thematic coding. Initial key themes for discussion were found: pricing method, solid waste integration, incorporation of reuse chain, scale, input waste control, market perception, standardisation and regulation.

*2.4. Ethics*

Where interviews were used to obtain values, this received ethical approval from Cranfield University Research Ethics, CURES/5687/2018.

**3. Results**

The results are divided into sections to answer the original research questions: 1. What is the financial value of FS re-use? 2. Is there a gap between theoretical value projections and the actual value that can be recovered? 3. What determines the value of FS reuse? 4. Can FS re-use act as a driver for improved sanitation? Section 3.1 presents an overview of the values of FS reuse from this review process. This refers to research questions 1 and 2 and provides an overview of the current value of FS reuse. Sections 3.2–3.7 explore the details behind the values and why there are differences in values, answering question 3. The major themes identified are: theoretical pricing, integration of organic solid waste, scale of operations, input waste availability, recognition and certification of products and integration of the supply chain. Understanding question 3 can help to inform whether the value of FS re-use could be improved by policy and scale, or whether there are limits to its potential. Understanding this enables the research to answer question 4.

*3.1. Overall Value of Faecal Sludge*

Whilst the theory scores give an indication of the data reliability (Table 2), it is also worth noting much of the literature available currently is grey non-academic literature; for example, from non-governmental organisations (NGOs), local governments or organisations active in the production and sale of waste-derived products [9,23,32] and, in general, are not commercial business operations devoid of subsidy. Based on observed production and sales of treated FS, only one value was above $5/person/year (p/y) (Figure 1). This value was based on co-composting of sludge with solid waste [54]: how the value was attributed to sludge rather than other waste streams (mainly organic solid waste) is unclear. Looking at the higher theory scores—those papers proposing an untested value proposition—the range of potential values increased up to $50/p/y. However, 72% of values were under $5/p/y, giving a good indication of the overall potential scale of resource recovery currently.

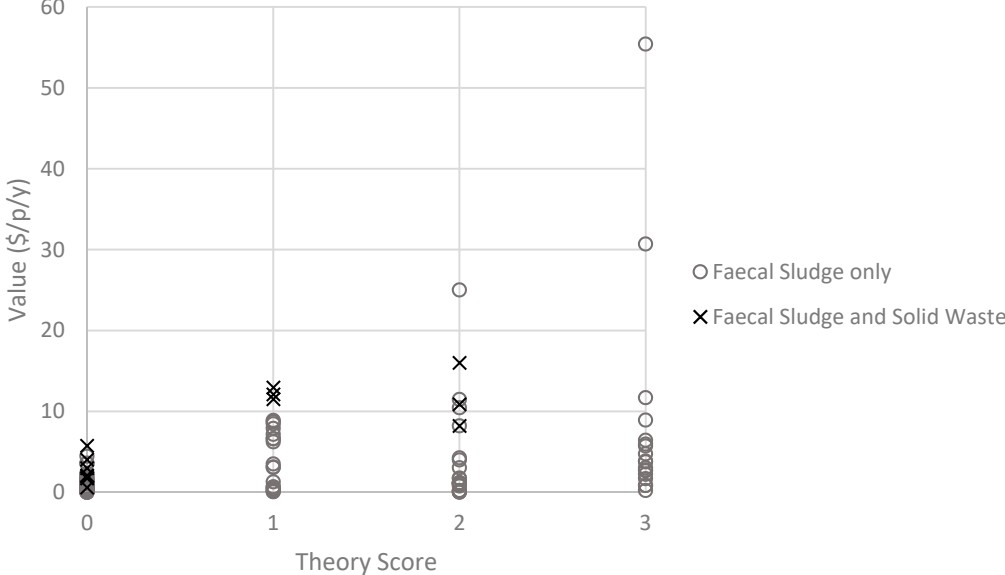

**Figure 1.** Value of faecal sludge reuse.

The most common form of reuse as researched and published is composting for agriculture, which is also the lowest value product [14,20,21] (Table 1). The agricultural reuse of excreta has a history going back thousands of years and is more technically understood and abundant as a reuse option than other more recent systems or resources [60,61]. The more recent forms of reuse, including Black Soldier Fly, may prove to be more profitable, but limited published research makes comparison difficult.

Taking Hutton's example of costing for the approach of achieving universal access to sanitation in South Asia, the reported cost of universal access was $10 billion for a population of 400 million, amounting to $25/p/y served [62]. However, this neglects the cost of maintaining existing services and ignores the potential contribution of recovered FS in reducing the costs of sanitation [62]. In Dhaka it was found that providing small sewerage systems cost $90–100 per household, per year, (with an average household consisting of 4.5 people), resulting in annual costs of sanitation of $20–23/p/y [63]. In this context, resource recovery on the scale of $5/p/y is unlikely to create a large contribution to funding the upstream sanitation chain. However, this also negates the cost of building and operating CE recovery systems and the business costs associated with marketing and sales of products, which are currently subsidised and funded by external organisations. The one paper covering the costs associated with the CE recovery element found additional costs accounted for 60% of the revenues from compost sales [34]. Overall, the examples of FS reuse in practice do not have a value above $5/p/y. There is currently a large gap between this and higher theoretical values reported by the literature. Sections 3.2–3.7 explore the determinants of this gap to assess whether it can be narrowed.

### 3.2. Theoretical Pricing

One major contributor to the high projected values for FS reuse was the method used to assign a price to the end-product. In order to see where the opportunities for shifting the potential of CE might lie, revenues above $5/p/y were studied, and it was found that many of these values were based on a pricing method that linked the product value of the FS to a commercially existing product. As an example, when pricing the potential of FS-based fuel, Diener et al. [14] found a potential production of 17 GJ/tonne. This was then pegged to a price per KJ of different fuels in Kampala, giving results ranging from $0.03 (coffee husks) to $23.20 (diesel), based on the product that FS-based fuel is expected to replace. Only diesel and engine oil replacement pegged prices gave a potential of over $5/p/y for FS reuse, and the results of this review indicate these seem the least likely markets for resource recovery to be contending for in low-income countries.

The potential value of FS-based fuel does also have other variables as prices for fuel are set and infrastructure funded, in some cases, by the national government, whereas biogas would require significant investment for starting large scale consumer use. If FS products could be sold at the same price/KJ as charcoal, it could collect revenues of $3.95/p/y, which is much more attractive than the coffee husk use ($0.03/p/y), which shows the potential variation in the price of end-products. This method of comparison pricing is also used in other papers looking at the potential of the sale of FS for fuel [21,64]. It is worth considering that these values will be highly contextual and based on local market conditions. For instance, crises such as the recent water shortages in Cape Town [65] could drive the value of models that recover water up. Similarly, in Ghana, frequent power cuts have provided a political need to diversify sources of electricity, which contributes to the drive there for biogas-to-electricity production [27].

One determinant of price that is difficult to assess from the results of this paper is competition in resource markets due to increasing resource scarcity or changes in financial markets or policy due to the unsustainability of certain methods of production. For example, China has currently imposed large export tariffs on phosphate rock, a key ingredient of chemical fertiliser, and the United States currently is estimated to have only 25 years of domestic reserves left [66]. The remaining global resources are in Morocco [66]. Furthermore, there can also be local price trends influencing the competing fertilisers. Policies such as those enacted in China, which aim to reduce chemical fertiliser usage and encourage organic fertiliser use, could shift the market realities for CE sanitation. Carbon credits and payment for ecosystem services have also been shown to shift revenue propositions for businesses producing organic fertilisers [8]. In India, the revenues and costs of compost production were about equal, making it an unattractive business venture, but subsidies provided at around $28/tonne enabled financial viability for some [53]. There were, however, smaller organisations trying to produce and sell compost who were not operating at sufficient scale to access a subsidy, and were subsequently unprofitable [13,53,54]. In Kenya, a sugar cane business used the residue to produce biogas for electricity and received around $270,000 of carbon credits, thereby covering around 27% of operating costs, showing the potential of subsidies and carbon credits to change the financial value of waste reuse [8].

The potential for FS reuse to produce large revenues often rests on the assumption that the price of the reuse product would be equal to the product it is replacing, such as chemical fertilisers or high-end fuel sources like diesel; however, these theoretical prices have rarely been achieved in actual market practice.

*3.3. Integration of Organic Solid Waste*

Whilst the literature cited often focused solely on the reuse of FS [14,20,27], upon further review of the data, a major driver of these businesses was combined integration of FS together with other organic waste streams (compostable solid waste). These waste streams are less hazardous to public health and thus can be more easily transformed for productive reuse than FS. Taking the example of resource recovery in Kampala, FS and sewage sludge accounted for less than 7% of the potential value recovered from energy, fertiliser or protein products [25], with organic solid waste accounting for the remaining potential value. Other business models, particularly for biogas, have solid waste integrated in the model but do not calculate the contribution from FS versus other organic waste streams [8,13,20,27,34,54]. Data points in this study with a value over $5/p/y, particularly from biogas, could be neglecting the role organic solid waste is playing in this system.

In interviews with three businesses producing different resources from FS, they reported a key constraint and determinant of their business model feasibility was the ability to access a reliable chain of input materials in addition to FS. This included either getting woodchip or charcoal for fuel briquettes, which could contribute up to 99% of the product, or organic solid waste for fertiliser and Black Soldier Fly larvae feed. The integration of organic waste streams has been cited as important in enabling CE sanitation [13,21,54,67], and the separation of municipal waste into organic and non-organic streams is going to be a critical step for scaling up CE sanitation.

### 3.4. Scale of Operations

The scale of operations is a major factor in the difference between the observed and the theoretical values (Figure 1), with theoretical value studies often focusing on whole cities or countries [14,21,25] whereas, in practice, current businesses are usually operating at a much smaller scale [5,6,8,34]. Whilst the scaling up of CE systems could theoretically help sanitation systems to break even [6], the increased production would need an expanded market of buyers. In Haiti, the sale of FS-derived compost depended on NGOs being willing to pay a premium price. Similarly, in Kenya, medium-scale horticultural farmers were willing to pay a premium price due to the product quality and targeted marketing [67]. Considering the model of whole city resource recovery, it is not clear whether there would still be a large enough market to buy compost at this premium price when scaled up. In Kenya, issues of transport costs to reach these market segments increased as enterprises needed to go further to find new customers [67]. Theoretical value projections based on waste from millions of people [21,22] may be unrealistic as the demand may not necessarily scale up easily with supply.

### 3.5. Input Waste Availability

The financial viability of any CE waste-derived business will depend on the quality and quantity of the input waste product. In a case in Ghana, existing businesses only took waste from public toilets [30], which are estimated to produce up to five times more biogas per kg of sludge than household septic tanks due to the lower retention time [68]. This means that businesses arranging to only take public toilet sludge are likely to be more financially effective than any city-wide solution. This control of input sources can also be seen in Container-Based Sanitation (CBS) businesses [5]. CBS is a system where waste is captured in sealable containers before transport to a treatment site, often with the collection of containers every one or two weeks [5]. The regular collection and small containers involved in CBS mean the waste collected is often a relatively standard input to the system and is less likely to be mixed with solid waste [69]. CE sanitation is heavily dependent on the availability of FS at a high quality and quantity: if the trash content is too high, or the FS is too difficult to remove, it is not financially viable.

### 3.6. Recognition and Certification of Products

Certification and recognition of product safety can improve the perception of value and products. In agriculture, a lack of regulation and certification of human-waste derived composts can lead to a lack of use of products, undermining their value [67]. In Peru, X-runner produces compost from FS but has not yet received permission from the government for sale thereof, meaning it effectively has no current financial value [43] as it is not approved for reuse. Other studies have shown that willingness to pay for FS derived products was influenced by whether or not the product was certified [70], such as in India where certification can also enable access to subsidies for fertiliser, making businesses more viable [13,53,54]. The highest sale prices from reuse business are found in India, where wholesalers who purchase compost are subsidised by $20.97 per tonne, which effectively inflates the sale price composters can get [13,53,54]. Looking at current the regulation of compost, the best example comes from Global Good Agricultural Practice, which does not permit use of FS in agriculture [71]. This means that human-waste derived composts cannot be adopted by farmers who are exporting produce, and this will inhibit both the global market development and ability to sell at premium prices [7]. In contrast, there are examples where households use untreated FS [8,11,13] despite its illegality and health risk.

### 3.7. Integration of Supply Chain

In India, compost production is often managed by a local utility, while a separate organisation takes on the role of marketing the fertiliser [13,56]. Whilst this means organisations with existing routes to farmers were in charge of selling compost, they also had limited interest in product promotion as they received more revenue from selling chemical fertilisers [53].

Integrating application of compost into the business model could increase the value of FS reuse. In systems where the compost value is incorporated and sold as vegetables [32,48,51], particularly from composting toilet designs, the estimated recovered revenue is on the scale of $100/p/y depending on crop and farm size. While this indicates a high value proposition of FS reuse in farming, shown in Table 3, these papers do not consider the other costs associated with growing crops. In the example of compost-producing companies, direct use and integration of farming into the business model may still reduce transport and marketing costs whilst producing a higher-value product. Sanergy's compost claims to improve yields by 30% on average for different vegetable crops compared to inorganic fertiliser alone [72], this estimates an increase in value of $987.15 per acre, whilst the compost sale price is $177.69–355.37 per acre (depending on application rates). It is already common for companies to compost waste from crops such as sugarcane and reuse it directly on subsequent crops [8].

**Table 3.** Reuse Values from Agricultural Activities.

| Source | Location and Crop | Size of Population | Value Per Person ($/p/y) |
|--------|-------------------|--------------------|--------------------------|
| [52] | India: banana | 700 | 9.97 |
| [51] | Uganda: apple | 6 | 320.14 |
| [32] | Kenya: mangos and bananas | 20 | 4.00 |
| [48] | South Africa: potato | 5 | 54.58 |
| | South Africa: maize | 5 | 253.32 |
| [40] | Burkina Faso: maize | 10 | 58.52 |

## 4. Discussion

This review suggests that a maximum value of products derived from FS is $5/p/y. In the one study where the operating costs along the whole chain from containment to reuse have been published along with reuse revenues, they only cover 10% of operating costs [34]. In another business practicing CBS reuse revenues only cover 8% of operating costs [6]. Higher value products often rely on smaller markets [6,34,43] or subsidies from government allowing the value to be effectively inflated [13,53,54]. This is unlikely to scale up when selling products from the treated waste of millions of people instead of tens of thousands, and no businesses are operating at that scale yet.

The main benefits of sanitation are hard to monetise, most specifically reduced health spending [62]. Resource recovery provides a new stream of revenue to mobilise businesses to enter into the market, but it still does not provide a huge financial incentive for organisations to start sanitation businesses or to invest in sanitation. For environmental, health and social reasons, CE systems of sanitation are worth pursuing, as they drive better waste management with all the associated health benefits. Unfortunately, most of these valuable contributions to society and the SDGs cannot currently be monetised. Carbon credits for electricity from biogas and subsidies for organic fertilisers are examples of interventions that help to shift the value proposition of CE sanitation. The state may need to intervene to create an enabling environment for CE sanitation [73].

### 4.1. Limits of the Study and Gap in the Literature

Limited data on resource recovery from FS treatment is available in low-income countries, and thus this work has considered both published and grey literature. Whilst many studies and grey literature are publishing the value of products from reuse, few report the costs of sanitation infrastructure and services at the same time, meaning this study has had to focus on only the value of reuse rather than the full costs and benefits. Further, this makes it hard to assess how much resource recovery could offset the overall costs of sanitation services. Where possible, this review has compared the potential revenue from reuse to available studies about the costs of sanitation services, but a need remains for studies publishing both the benefits of resource recovery and the costs of infrastructure in the same location to inform discussions about the scale of funding and the subsidies required. The presence of hazardous materials in FS, such as heavy metals and pathogens, can limit reuse potential and, hence,

value. There are examples in three different countries where CE actually functions as a profitable business with direct reuse with limited treatment, which presents high health risks for farm workers and crop consumers [11,13,54].

It is also hard to determine whether the higher value propositions are simply from economies where prices are higher for all goods, rather than just for the FS reuse products. Using measures such as purchasing power parity (PPP) exchange rates instead of market currency exchange rates could help to understand the variation amongst countries. Two further methodological issues remain with the literature reviewed in this paper. Firstly, studies often used prices in non-local currencies that are different to those used in the research, such as dollars, euros or pounds, particularly in more theoretical studies [14,25]. This means that disentangling values and understanding the original local context from which they were projected was not possible. Secondly, existing papers on the cost of sanitation used the market exchange rate rather than PPP [15,62,63], meaning data on resource recovery is more relevant if market exchange rates are used. How taxation applies to sanitation business was not considered, and this could also be a deterrent to CE sanitation success.

### 4.2. Future Outlooks in This Field

An important task in further research would be to assess the costs of sanitation systems in different contexts. A standardised protocol to investigate costs, with consideration of variations in international prices and local resource markets is being proposed by the Climate Costs in Urban Sanitation (CACTUS) project [74], and once data are available it would be interesting to reanalyse the findings that form this review. Further research into an economic analysis of external costs and benefits could shift the policy debate on supporting CE sanitation. The subsidies for compost in India could be seen as an example of a monetary representation of these externalities [13,53,54].

The majority of current research looks at agricultural reuse of FS. As more novel forms of reuse, such as Black Soldier Fly larvae, are developed, the financial value could become more promising, assuming there are markets for the products generated. The value of products can often be defined by locally available competitors, such as charcoal for fuel or chemical fertilisers, or lack of cash for these as imported supplies, and this dynamic is currently poorly understood.

## 5. Conclusions

This research looked at the economic potential of resource recovery from FS treatment in low-income countries. Sixty-five percent of the existing literature was based on projections rather than actual experiences of revenue in practice. Overall, in practice, FS reuse is unlikely to have a value above $5/p/y, and even these values often relied on subsidies or small markets that would not scale. There was a gap between the values presented in proposition papers, which suggested potential values of up to $50/p/y, and those that were observed data from operating businesses, which were up to $5/p/y. Reasons for this gap between practice and theory include the pricing of products against unrealistic competitors, such as pricing briquettes against diesel fuel, or difficulties in marketing or regulation of the products in practice. The review shows that the promise of CE sanitation is over emphasised in the current literature and is unlikely to act as a driver for improved sanitation. Overall, this research identified a lack of systematic data collection for sanitation costs in general and for the potential of resource recovery. Further research is needed to quantify the costs of sanitation provision and FS treatment, but it is unlikely that the value of reuse products will drive large improvements in sanitation because the main benefits are not easily monetised. Overall, it seems that CE sanitation and sales of products are unlikely to make a large contribution to the provision of sanitation in low-income countries overall, and, more importantly, sanitation cannot be achieved solely as a market approach. Instead, focus should be on supportive policies and investment, enabling the transition towards a CE supporting environmental quality, ecological health and human health to achieve SDG 6.

**Supplementary Materials:** The data from the literature review used to assess the value of faecal sludge reuse is available open access through the Cranfield Online Research Data repository at https://doi.org/10.17862/cranfield. rd.11336792.

**Author Contributions:** Methodology, funding acquisition, field work, modelling, data analysis, writing—original draft preparation, A.M.; conceptualisation, methodology, funding acquisition, supervision, project administration, writing—review and editing, A.P.; writing—review and editing, R.H. All authors have read and agreed to the published version of the manuscript.

**Funding:** This research was funded by the Royal Society CHL\R1\180402.

**Conflicts of Interest:** The authors declare no conflict of interest.

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
