# Peer review of "A Review of the Financial Value of Faecal Sludge Reuse in Low-Income Countries"

_sustainability, doi:10.3390/su12208334_

Round 1

Reviewer 1 Report

The revised manuscript versions responding effectively and competently to the reviewers(s) feedback, which is appreciated Here are my comments for the manuscript  improvement.

Introduction. The introduction could do more to ground the paper’s RQ in the debate and the related literature. In the actual version of the manuscript, scant attention is given to a theoretical derivation of the study’s RQ, and its actual positioning. Indeed, the lack of studies cannot be considered sufficient motivation for this study.Consider expand the introduction by highlighting the research gap and focus on what CE.

My suggestion is also to fully rewrite this section to answer the following questions: (i) Why is this topic relevant and what is known about it? (ii) Which are the gaps you plan to address and how do you problematize about them? (iii) How do you plan to close/address those gaps? (iv) Which are the main contributions of your study?

Literature review needs to be strengthened. Literature Review has the chance to be further improved: it seems that the authors have made the retrospection. However, via the review, what issues should be addressed? What is the current specific knowledge gap? What implication can be referred to? The above questions should be answered. Authors need to propose their own study

Results and conclusion. The section devoted to the explanation of the results suffers from the same problems revealed so far. Your storyline in the results section (and conclusion) is hard to follow. Moreover, the conclusions reached are far from what one can infer from the empirical results. The discussion should be rather organized around arguments avoiding simply describing details without providing much meaning. A real discussion should also link the findings of the study to theory and/or literature.

Minor General Comments

- The manuscript is potentially original contributive but needs a major revision.

-Some sentences from conclusion could be moved up in discussion section. Conclusion section must be well written and clearly explain study findings.

- Implications for future research may also be included in the conclusion at the end. This research has article has created a lively discussion on so many issues that were hitherto unheard of and not addressed.

- Also explain briefly what the future research opportunities are.

Buyer-Driven Knowledge Transfer Activities to Enhance Organizational Sustainability of Suppliers, Sustainability 2020, 12(7), 2993; https://doi.org/10.3390/su12072993.(Scopus).

Progress from blue world to green world: Multilevel governance for pollution prevention planning and sustainability. C. M. Hussain (ed.), Handbook of Environmental Materials Management, ISBN: 978-3-319-58538-3. https://doi.org/10.1007/978-3-319-58538-3_177-1.(Scopus).

Industrial ecology in support of sustainable development goals. Encyclopedia of the UN Sustainable Development Goals (UNSDGs), SDG 12, Springer Nature Switzerland AG.ISSN: 978-3-319-71062-4. Springer International Publishing, 1–12. DOI: https://doi.org/10.1007/978-3-319-71062-4. 978-3-319-71062-4.(Scopus).

Exploring the locus of social sustainability implementation: A South Asian Perspective on Planning for Sustainable. Universities and Sustainable Communities: Meeting the Goals of the Agenda 2030.World Sustainability Series, ISBN:978-3-030-30305-1. Springer Nature. https://doi.org/10.1007/978-3-030-30306-8 (Book).(Scopus).

Effects of the buyer-supplier relationship on social performance improvement and innovation performance improvement, International. Journal of Applied Management Science, Vol. 11, No. 1.pp.  21–35 (Q3, SCOPUS). Usama Awan (2019).

Governing inter-firm relationship for sustainability: The relationship between governance mechanism, sustainable collaboration and cultural intelligence, Sustainability,10(12). pp.1-12 (Q2, Scopus, Web of Science).

Reviewer 2 Report

Dear Authors,

The paper 'A review of the financial value of faecal sludge reuse in low-income countries" is an interesting review.  

The abstract needs to be rewritten, it does not provide essential intormation on the topic and the main findings of the review. It seems that the review fails to be carried critically. 

The keywords are a good representative of the text. 

The general introduction is acceptable, however the state of the art needs a serious rewriting, as it fails to be caried critically and it does not gefine a gap, purpose for the review to be written, that meets your goals. Then in the conclusions section it needs to be adressed. It would be more concise.

You should also consider to resent your scientificic hypothesis and procedure as a block diagram, analogical to DOE for research paper.

The clue for dataset choice needs to be clearly explained, it is very easy to manipulate the discussion, if certain datasets are chosen, please explain. also data in tables and diagrams should be provided with SD. 

Legal regulations review should be taken into account when considering compost as value-added product, as in different countries it may be a olytical issue.

The conclusions are concise, however they should be suorted by quantitave analysis or literature data review.

Reviewer 3 Report

The authors summarized 43 related paper to explore the current potential for the resource recovery of fecal sludge treatment in low-income countries. The authors analyzed the results from different themes and pointed out the gap in all those researches and pointed out the future research directions. The results analyses are comprehensive and the discussions are clear and insightful. I would recommend accepting the pare at the present form.  

Author Response

Many thanks for the summary and comments!

Round 2

Reviewer 1 Report

Author/s have carefully addressed all the comments. This paper is interesting, and it deals with a topic central to the field of research, however, several issues have not been adequately addressed. It is especially problematic in the introduction that brings up a number of findings from different areas without linking them together. Furthermore, the study touches upon several large research areas. There are inconsistencies and several concepts, which have not been explained. In the introduction, you need to connect the state of the art to your paper goals. Currently, this is not performed in a convincing way. Please follow the literature review by a clear and concise state of the art analysis. This should clearly show the knowledge gaps identified and link them to your paper goals. Please reason both the novelty and the relevance of your paper goals. The methodological part is neither well written nor explained and informally written. The results appear weak, and the methodology is not appropriate. There are a few specific issues the authors should address by making modifications to the manuscript or by clarifying in their response, after which I would recommend this work suitable for publication

Reviewer 2 Report

Dear Authors, thank you for addressing the comments, although not many significant changes were introduced in the text, I do understand the intentions. 

Author Response

Many thanks for noting our changes.

Round 3

Reviewer 1 Report

However, while covering broad topics and different types of studies, this approach holds that the research process should be transparent and should have a developed research strategy that enables readers to assess whether the arguments for the judgments made were reasonable, both for the chosen topic and from a methodological perspective. Again, I suggest authors to improve the paper by taking into consideration following suggestions. Authors have addressed some of these issues in manuscript, but  I suggest authors need to explain bit more with appropriate literature.

- Authors can present the research gaps/issues in a separate sub-section at the end of the literature review and also provide more emphasis on how the research gaps are bridged in the paper by highlighting the original contribution to the paper.

- For readers to quickly catch the contribution in this work, it would be better to highlight major difficulties and challenges, and your original achievements to overcome them, in a clearer way in abstract and introduction.

In terms of research quality, deciding on inclusion and exclusion criteria is one of the most important steps when conducting your review. However, important to note is the need to provide reasoning and transparency concerning all choices made; there must be logical and valid motives.I also suggest to expand methodology part , why review studies are important and what are its strengths and  weaknesses.

-Major areas of agreement and disagreement in the literature should be discussed. The discussion should tie the study into the current body of literature, provide its significance, and make logical interpretations from the literature review

Author Response

As authors we believe that there is nothing valuable in this review.  Not only does the reviewer give us irrelevant comments that do not pay attention to our manuscript or previous response, we now also believe that the review itself is plagiarized.

The following statement is copied from Snyder (2019):

In terms of research quality, deciding on inclusion and exclusion criteria is one of the most important steps when conducting your review. However, important to note is the need to provide reasoning and transparency concerning all choices made; there must be logical and valid motives.I also suggest to expand methodology part , why review studies are important and what are its strengths and  weaknesses.

The following statement is copied from Green et al (2006):

Major areas of agreement and disagreement in the literature should be discussed. The discussion should tie the study into the current body of literature, provide its significance, and make logical interpretations from the literature review.

As such we have made no further changes to our manuscript.

Green BN, Johnson CD, Adams A. Writing narrative literature reviews for peer-reviewed journals: secrets of the trade. J Sports Chiropr Rehabil 2001;15:5–19.

Snyder (2019) Literature review as a research methodology: An overview and guidelines, Journal of Business Research 104 (2019) 333–339

This manuscript is a resubmission of an earlier submission. The following is a list of the peer review reports and author responses from that submission.

Round 1

Reviewer 1 Report

The topic of the paper is potentially interesting. However, I think there are some critical points and weaknesses that impede to publish the article

- The abstract needs a complete reformulation. The current version is a half-page text that first repeats the title and then says nothing of essence. This has to be corrected. Please provide a meaningful abstract - follow the same pattern as recommended for the highlights, but in smooth text.

- The introduction section is poorly organized. While the general introduction is acceptable, the state of the art review that follows is very difficult to understand and no specific thoughts can be inferred. This part needs a serious re-write.

- In the introduction, you need to connect the state of the art to your paper goals. Currently, this is not performed in a convincing way. Please follow the literature review by a clear and concise state of the art analysis. This should clearly show the knowledge gaps identified and link them to your paper goals. Please reason both the novelty and the relevance of your paper goals.

- Before proceeding to describe your chosen model and actions, please describe your scientific hypothesis, concepts and the relevant reasoning for choosing the particular approach. This should be accompanied by an overall description of the followed procedure. A block diagram of the procedure would be also very useful

- The conclusion also needs improvement. In the conclusion, in addition to summarizing the actions taken and results, please do explain their significance. It is recommended to use quantitative reasoning comparing with appropriate benchmarks, especially those stemming from previous work. While this study presents a specific problem area, nevertheless, the improvement to the state of the art has to be clearly shown and demonstrated by the results and stressed in the conclusion.

Reviewer 2 Report

This manuscript review of the financial value of faecal sludge reuse in low-income countries. Some of issues should be addressed.

  1. The significant findings of the manuscript should be addressed and elaborated.
  2. Data collected or analyzed in this study should be provided and presented in either figures or tables (including raw data).

Reviewer 3 Report

Dear Authors, the paper 'A review of the financial value of faecal sludge reuse in low-income countries" is an interesting REVIEW with many up-to date findings. 

The aabstract and keywords are a good representative of the text. 

The introduction provides a good bacground on the field, however I would suggest to update the reference list, to highlight the importance of carried discussion. Moreover, the Authors should mention the main aim and then show how these was met in the concluding section. It would be more concise.

The clue for dataset choice needs to be clearly explained, it is very easy to manipulate the discussion, if certain datasets are chosen, please explain.

Data in table 3 should be provided with SD or at least period should be clearly defined in the table caption.

The sentence "It is already common for companies to compost waste from crops like sugar and reuse it directly on subsequent crops" needs to be rewritten, the food vs. compost debate is not as clear, sugar is not a crop.

also legal regulation should be taken into account when considering compost as value-added product.

The results are generally supporting the conclusions.

The conclusions are concise, maybe add a few bullet points regarding the contribution of yoour approach to the field